# Gender disparities in social and personality psychology awards from 1968 to 2021
Aífe Hopkins-Doyle [1] ✉, Jocelyn Chalmers [2], Daniel Toribio-Flórez [2] & Aleksandra Cichocka [2]

Gender disparities persist in academic psychology. The present study extended previous investigations to social and personality psychology award recipients. We collated publicly available data on award winners ($N = 2700$) from 17 international societies from 1968 to 2021. Features of the award, including year given, type of award, seniority level, whether the award was shared with more than one winner, and gender/sex of the recipient were coded. Overall, men were more likely to be recognized with awards than women, but the proportion of awards won by women has increased over time. Despite this increased share of awards, women were more likely to win awards for service and teaching (which are generally viewed as less prestigious) rather than research contributions. These differences were moderated by year - women were more likely to win service or teaching awards, compared to research awards, after 1999 and 2007, respectively. Women were more likely to win awards at postgraduate/early career levels or open to all levels compared to senior awards. Findings suggest that women's greater representation in academic psychology in recent years has not been accompanied by parity in professional recognition and eminence.

In academia, women are underrepresented in almost all scientific disciplines, hold fewer senior and prestigious positions, receive a lower proportion of grant funding, and are paid less than their colleagues who are men[1–3]. Women's research excellence and scientific contributions are also recognized less often with awards[4–6]. Yet, awards are important for career development – highlighting excellence to funding agencies, tenure, and promotion committees[7]. Beyond individual benefits, awards help shape disciplines by indicating which contributions are valued and should be aspired to[8–10]. In psychological sciences, women are not underrepresented numerically, but still experience gender/sex disparities in publication rates, citations, and pay gaps[11–14]. We investigated whether this pattern extends to awards recognition in social and personality psychology – a subdiscipline which, among other topics, focuses on understanding prejudice and bias in decision making. We analyzed publicly available data of awards from 17 international social and personality psychology societies from 1968 to 2021. We examined whether features of the award (e.g., award type, seniority level) were associated with gender/sex disparities and, importantly, whether patterns have changed over time.

Decades of research on gender stereotyping offer theoretical reasons to expect gender/sex disparities in awards. Gender stereotypes describe how women and men are, distinguishing them in terms of communality (i.e., other-orientated traits) and agency (i.e., self-orientated traits[15]). Women are typically seen as possessing communal traits such as warmth, helpfulness, and trustworthiness, whereas men are seen as having agentic attributes including independence, ambition, and competitiveness[16]. Gender stereotypes create expectancies about individual performance, even if we are not consciously aware of this. They tend to be consensual across cultures[17] and relatively stable over time[18], although recent research suggests a shift towards greater gender egalitarianism at an implicit and explicit level[18,19].

Performance expectancies derived from gender stereotypes have important consequences for women's workplace progression. The lack of fit model[16,20] proposes that difficulties arise when expectations of the environment do not align with descriptive stereotypes of women. For masculine-typed occupations, such as those in science, technology, engineering, and mathematics (STEM), agentic traits are prescribed which are seen as at odds with women's communality – this mismatch reduces perceived suitability, and expectations of success in the role. Numerous studies – using field data, experimental, and meta-analytic approaches – demonstrate the influence of stereotyped expectancies in career progression[21–23]. Further, there is some evidence that negative expectancies influence how women's contributions in work are valued. Women working in dyads with men are seen as less responsible for the success of a joint task, and this was only attenuated when individual performance information was explicitly given[24]. In fact, this tendency to overlook women's scientific contributions and view them as less

[1]Department of Psychological Sciences, University of Surrey, Guildford, UK. [2]School of Psychology, University of Kent, Canterbury, UK. ✉e-mail: a.hopkins-doyle@surrey.ac.uk

important than men's is well-documented enough that Rossiter[25] coined the term the Matilda Effect to call attention to it. This idea is pertinent in the context of evaluating contributions to scientific excellence which is often achieved through collaboration.

Beyond women's greater communality and lower agency, stereotypes about their intellectual capacities may also influence evaluation of their work. Research shows that men are more likely than women to be associated with intellectual brilliance or genius both explicitly and implicitly[26,27]. In fact, the number of women in a given academic discipline is negatively related to the expectation that brilliance is necessary for success in that discipline[28] (for alternative interpretation see Ginther and Kahn[29]). Further, experimental research shows women (versus men) candidates are 19.9% less likely to be referred for a job when intellectual brilliance is included in the job description[30]. Given this lower ascription of innate genius, it may be easier to overlook women than men when it comes to awards. This could be particularly important for some types of awards, such as those explicitly given for single pieces of brilliant work (e.g., Wegner Theoretical Innovation Prize from the Society for Personality and Social Psychology).

Gender stereotypes also influence how individual women assess their own ability (i.e., self-stereotyping) and the roles that they pursue. Perceptions of stereotype-consistent attributes can lead women to self-select into roles which are congruent with their gender role[31,32]. For example, women are more likely than men to occupy teaching-intensive positions that make it more difficult to pursue research goals, and may also be granted less agency and status than men within research departments[33]. This role segregation within academic departments means women are less likely to meet thresholds in terms of citations and *h*-indices to win awards for research contributions[34]. At the same time, it might make women more competitive for awards given for teaching or service contributions to the field.

Notwithstanding theoretical reasons for gender disparities, there are also structural changes over time that might impact who receives professional recognition. For example, at least in US psychology, women now outnumber men in PhD conferrals, professional society membership, and junior faculty[35–37]. Despite these advances, women are still underrepresented at top faculty levels (e.g., professors, deans) and decision making positions (e.g., editors, society presidents) compared to men[35,37,38]. Beyond the US, analyses of the European Association of Social Psychology (EASP) membership show women make up equal numbers of members but are likewise underrepresented in prestigious roles including symposia at the general meeting, and in the most prestigious role of the society (only two of 19 Presidents have been women[39]). One potential consequence of women's continued underrepresentation at top levels is that their contributions may not meet the perceived eminence and status thresholds for the most senior honors (i.e., lifetime achievement awards[12,37]). We might expect then that women would be less likely to receive awards at senior levels compared to men, and perhaps are more likely to win awards at junior levels.

Research from STEM fields shows that gender disparities exist in award giving but are more or less pronounced depending on the year given, type, and level of the award. Analysis by Lincoln et al.[6] of award recipients across 13 STEM societies (between 1991 and 2010) showed that, over time, women won more awards overall, and their share of award wins for research (i.e., the most prestigious award type) was roughly proportional to their numbers in the wider field, but men were twice as likely as women to win research awards regardless of their representation in the pool of nominees. A separate analysis of the most prestigious international science research awards given between 2001 and 2020 showed a similar pattern[7]. Over time, women have increased their share of prestigious international awards from 6% in 2001-2005, to 19% in 2016-2020, but these figures are in stark contrast to the number of women holding professorships during these periods, 17% and 28% respectively. There is also evidence that women win different types of awards than men. Analyses of biomedicine prizes showed women were more likely to win for service or teaching rather than research contributions, and for every prize dollar a man received, a woman got 64.4 cents[4].

The limited existing research into gender disparities in psychology awards paints a similar picture. In 2017, the American Psychological Association (APA) reported that women have never received more than 38% of its awards, and this proportion is considerably reduced when examining more prestigious and senior awards (e.g., only 15% of Distinguished Scientific Contributions)[35]. An extended investigation of APA awards by Orchowski and colleagues[40], shows that overall women received fewer awards than men (27.4% women, 1956–2019), but that their share did increase slightly between 1999 and 2019 to 35.1%. Of the 10 recognition awards investigated, it is notable that women only surpassed or came close to parity with men in two categories (i.e., Research in Public Policy, 56.3% women; and Humanitarianism, 43.8% women) both of which are consistent with communal stereotypes of women's greater morality and concern for others[18]. Investigations of gender disparities beyond the US that specifically focused on social and personality psychology are limited to a single society - EASP[39]. Consistent with the APA study by Orchowski[40], analyses showed disparities emerging at more senior levels and continuing over time. Specifically, while the share of more junior awards (i.e., Early Career Best Manuscript) won by women reflects the numbers of postgraduate women in the society (71.4% versus 71.8%), men outstripped women in total awards won since 1984, and only three out of 13 winners of the most senior EASP award (formerly named the Tajfel medal) were women. Overall, these findings highlight slow progress for women in the recognition of their scientific contributions despite their majority status as members of the APA or EASP. However, they are limited by a focus on a few awards in particular societies and contexts. They also did not examine systematically how gender disparities in psychology awards might have changed over time.

The aim of the present study was to investigate potential gender disparities in the subdiscipline of social and personality psychology. We chose social and personality psychology because of the field's focus on investigating potential prejudices in behavior and decision making and because, unlike other subdisciplines, it is relatively gender/sex balanced in numbers of PhD graduates, faculty, and society members[35,39]. We also investigated change over time in who wins awards and tested whether year award was given interacted with other features of the award (i.e., type, seniority level, sharedness) in predicting gender disparities. To do this, we reviewed and collated award information from 17 well known international professional societies between 1968 and 2021 (see Table 1 for details of these societies). Based on social psychological theories of gender stereotyping and roles[16,35], we expected that women would receive fewer awards overall compared to men, but that they may be more likely to win awards seen as congruent with communal gender roles (e.g., teaching and service) compared to research awards. In line with the lack of fit model, women's contributions regardless of quality, may still be valued less, and therefore we expect women to be more likely to win an award in conjunction with others each year compared to men. Finally, in respect of our focal investigation of change over time and the potential moderating effects of time, we did not have any theoretical bases for predictions, however given increasing numbers of women in the discipline and trends from STEM more broadly, it is likely that women would win more awards over time. We sought to explore moderating effects of time.

## Methods

First, we identified professional societies in social and personality psychology, and the awards given by each organization. We aimed to include international social and personality psychology societies which had award winners publicly listed on their websites. We decided not to include national societies because we did not identify a comprehensive list of such organizations and because many of them do not have publicly accessible lists of awards presented in English. Nine societies were generated initially through discussion between the first and last author. Following this, we undertook a further search for relevant societies using those listed on the Psychology Organizations and Conferences website (socialpsychology.org). Doing so identified a further four societies for inclusion. Lastly, a general internet search identified an additional four societies mostly focused on personality and cultural psychology. See superscripts in Table 1 for details.

**Table 1 | Awards given by each society in total, to women, and to men**

| Society | Acronym | Total awards given n(%) | Awards given to Women n(%) | Awards given to Men n(%) |
|---|---|---|---|---|
| Asian Association of Social Psychology[b] | AASP | 109 (4.0) | 47 (43.1) | 62 (56.9) |
| American Arab, Middle Eastern, and North African Psychological Association[c] | AMENA-PSY | 9 (0.3) | 9 (100) | 0 |
| Association for Research in Personality[b] | ARP | 99 (3.7) | 34 (34.3) | 65 (65.7) |
| European Association of Personality Psychology[c] | EAPP | 98 (3.6) | 31 (31.6) | 67 (68.4) |
| European Association of Social Psychology[a] | EASP | 80 (3.0) | 27 (33.8) | 51 (63.7) |
| International Association for Cross Cultural Psychology[a] | IACCP | 19 (0.7) | 11 (57.9) | 8 (42.1) |
| International Social Cognition Network[a] | ISCON | 71 (2.6) | 16 (22.5) | 55 (77.5) |
| International Society for Political Psychology[a] | ISPP | 265 (9.8) | 89 (33.6) | 173 (65.3) |
| International Society for Self and Identity[a] | ISSI | 85 (3.1) | 34 (40.0) | 51 (60.0) |
| International Society for the Study of Individual Differences[c] | ISSID | 35 (1.3) | 7 (20.0) | 28 (80.0) |
| Society of Australasian Social Psychologists[a] | SASP | 14 (0.5) | 6 (42.9) | 8 (57.1) |
| Society of Experimental Social Psychology[a] | SESP | 259 (9.6) | 95 (36.7) | 164 (63.3) |
| Society for the Psychological Study of Social Issues[a] | SPSSI | 823 (30.5) | 417 (50.7) | 406 (49.3) |
| Society for Personality and Social Psychology[a] | SPSP | 535 (19.8) | 194 (36.3) | 338 (63.2) |
| Society for Social Neuroscience[b] | S4SN | 19 (0.7) | 10 (52.6) | 9 (47.4) |
| Society for the Psychological Study of Race, Ethnicity and Culture[b] | SPREC | 160 (5.9) | 85 (53.1) | 75 (46.9) |
| World Association for Personality Psychology[c] | WAPP | 20 (0.7) | 9 (45.0) | 11 (55.0) |
| Total | | 2700 (100) | 1121 (41.5) | 1571 (58.2) |

A total of $n = 8$ (0.3%) cases for gender/sex could not be categorized. These were for EASP ($n = 2$), ISPP ($n = 3$) and SPSP ($n = 3$). These uncategorized awards represented 2.5%, 1.1%, and 0.6% of the total awards given within each society. [a]Societies identified first during discussion by authors. [b]Societies identified via socialpsychology.org list. [c]Societies identified by general internet search.

Next, we manually collated information on winners of each award, in each year, for each society. As there were no human participants, no software was used to collect the data. We gathered information on the name of the award, the description of the award (as specified by the society), and the name(s) of winner(s) each year. To maximize the number of data points, we decided not to restrict the range of years in which the award was given, but to collect information on awards from all available years. Data was collected between 2020 and 2021, thus this was our upper limit for year. Next, one of the authors coded the data gathered using the following categories: gender/sex presentation of the award winner, award type, award level, shared awards, and honorable mentions. We use the term gender/sex to acknowledge the interconnected and often inseparability of sociocultural and biological features of gender and sex (see ref. 41). This is particularly pertinent for the current investigation given the limited information available to code the awardees' gender/sex (i.e., given names, pronouns, clothing, hair style, make-up etc.). We also coded whether awards had changed from a shared award or not shared award in the last cycle (labelled *procedural change in award*, although this variable is not analyzed). See Table 2 for further details of variables coded including criteria, sources of information, coding, definitions and examples, and frequencies. Ethical approval was provided by the School of Psychology Ethics Committee at the University of Kent, UK. The study, including our predictions, were not pre-registered.

The coding of all categories was made on the basis of publicly available information on society websites (e.g., descriptions of the awards, announcements of winners) and for winners' gender/sex we also conducted an online search of their professional webpage, university profile, or other professional online presence. If it was not possible to identify gender/sex through these means, categorization was made by entering the awardee's first name into a marketing tool Gender API, which identifies the typical gender/sex associated with a first name (for analysis of accuracy see Santamaría and Mihaljević[42]). For award type, if an award description mentioned research in addition to one of the other category types, it was coded

exclusively as research. For honorable mentions, we coded gender/sex in the same way as winners in that year.

**Statistics and reproducibility**

The dataset includes 2700 observations. The $n$ varies slightly between tests because we excluded some categories (e.g., "not applicable" category for gender/sex or award level variable) or there was a small number of missing data for year ($n = 4$). All these cases were from a single society (SESP) and listed year as "pre-1977" on the society website. We decided to leave year blank in these instances. An alpha level of 0.05 for statistical significance was used. SPSS (version 28/29) was used for analyses, which were reproduced in R version 4.3.

A one-sample chi-square test was used to compare observed and expected frequencies at which men and women received awards ($n = 2692$). To examine associations between gender/sex and each feature of the award we used Pearson's Chi-Square Test of Independence. Assumptions of Chi-Square were checked and met. Gender/sex of the winner was recoded (0 = man, 1 = woman) for ease of interpretation. For Chi-square the effect size is Cohen's omega ($\omega$), *Cramer's V* ($\varphi_c$) or *odds ratio* (OR). Cohen's omega and Cramer's V indicate the strength of association from 0 to 1, where higher scores indicate a stronger association. Odds ratio is only applicable for 2×2 contingency tables and indicates the likelihood of an event happening compared to not happening. In this case, if OR is > 1 it indicates women are more likely to win than men, and if OR is < 1 it indicates that women are less likely to win than men.

Sample size was $n = 2692$ for award type, shared, and honorable mention. Note that for these analyses the gender/sex "not applicable" ($n = 8$) category was excluded. For award level, the sample size was $n = 2633$ as there was a total of $n = 67$ cases excluded which were categorized as "not applicable" for award level ($n = 63$), or "not applicable" for gender/sex ($n = 4$). For award level and type, we ran follow up sub-sample analyses to breakdown the significant Chi-square test. We dummy coded each award type and level into a binary variable comparing the category of interest (equal to 1) with all other categories (equal to 0). For shared awards,

**Table 2 | All variables coded on social and personality psychology award winners and their frequencies**

| Variable | Criteria | Sources of information | Definition and Example | Coding | N (%) |
|---|---|---|---|---|---|
| Gender/Sex Presentation of Award Winner | Whether the winner was a woman or a man? | • Use of gendered pronouns in biography descriptions or CVs. • Photos of the individual which indicated a binary gendered appearance. • Given name by entering it into the Gender API. | • Use of she/her; stereotypical woman's appearance (e.g., long hair, make-up). | • Woman = 1 | 1121 (41.5) |
| | | | • Use of he/him; stereotypical man's appearance (e.g., short hair, facial hair). | • Man = 2 | 1571 (58.2) |
| | | | - | • Unknown = 3 | 8 (0.3) |
| Award Type | What type of contribution the award was given for? | • Society website. • Descriptions of the award. | • Given for research contribution e.g., Cialdini Award, SPSP; Murray Award, ARP. | • Research = 1 | 2127 (78.8) |
| | | | • Impact beyond academia, practical applications of psychology and public engagement with science e.g., Nevitt Sanford Award, ISPP. | • Impact/media = 2 | 136 (5) |
| | | | • Contribution to the development of the goals and reputation of the discipline/specific society, e.g., Service to SPSP Award, SPSP. | • Service = 3 | 253 (9.4) |
| | | | • Contributions to teaching excellence including student mentorship e.g., Innovative Teaching Award, SPSSI; Charles and Shirley Thomas Award, SPREC. | • Teaching = 4 | 184 (6.8) |
| Award Level | What career stage the award was open to? | • Society website. • Descriptions of the award. | • Professorial awards, lifetime achievement awards. | • Senior = 1 | 771 (28.6) |
| | | | • Open to those 10 – 25 years post PhD. | • Mid-career = 2 | 69 (2.6) |
| | | | • Open to those with a doctoral degree. | • Early career researcher (ECR) = 3 | 727 (26.9)[a] |
| | | | • Open to those currently undertaking doctoral studies. | • Postgraduate (PG) = 4 | |
| | | | • Open to both doctoral and post-doctoral levels. | • Postgraduate/early career researcher (PG/ECR) = 5 | |
| | | | • Open to all levels of experience. | • All Levels = 6 | 1070 (39.6) |
| | | | • Non-academic winners e.g., media figures. | • Not applicable = 0 | 63 (2.3) |
| Shared Awards | Whether the award was shared or not between two or more winners each year? | • Society website. | • A single winner each year. | • Not shared = 0 | 934 (34.6) |
| | | | • More than one winner each year. | • Shared = 1 | 1766 (65.4) |
| Honorable Mentions | Whether the award included honorable mentions each year or not? | • Society website. | • No honorable mention that year. | • No honorable mention = 0 | 2421 (89.7) |
| | | | • Honorable mention that year. | • Honorable mention = 1 | 279 (10.3) |

[a]These estimates reflect all ECR and PG categories in aggregate as we collapsed across these categories for analyses. Breakdowns for each category are: ECR ($n$ = 427, 15.8%), PG ($n$ = 265, 9.8%), PG/ECR ($n$ = 35, 1.3%).

honorable mentions, and sub-sample analyses of award type and level Yates' continuity correction was reported.

For our main model we ran a logistic regression ($n = 2629$) to investigate the joint associations between the different award features and gender/sex of the winner. Gender/sex of the winner was recoded (0 = man, 1 = woman) for ease of interpretation. Assumptions of logistic regression were checked and met. In logistic regression, the effect size is OR. Here it indicates the likelihood of the award winner being a woman over a man. Confidence intervals (95%) are reported for ORs. For categorical variables, award type and award level, we created dummy variables to compare each category with a reference (or baseline) category. For award type, "research" was entered as the reference group. For award level, "senior" was entered as the reference group. Note that the "non-applicable" group for award level was excluded from this analysis as these cases represented non-academic winners. As the data are clustered within societies, we reproduced this analysis using a multilevel framework in R. Results were very similar to those reported in the results section (see Supplementary Table 1 and Supplementary Fig. 1 for full details).

We ran an additional logistic regression to investigate whether award features (type, level, shared, and honorable mentions) were moderated by year ($n = 2629$). The same exclusion of categories applied to this model. To further probe significant interaction effects, we reran this model using the Johnson-Neyman technique in the Process macro for SPSS (Model 1, version 4.2[43]). The Johnson-Neyman technique identifies a cut-off point on the moderator at which the effect of the predictor on the outcome is significant[44]. In our case, this means the exact year at which the effect of award type on winner's gender/sex becomes significant. In both models ($n = 2629$), year was entered as the moderator (W), and winner gender/sex as outcome (Y). In the first model, the service dummy (research = 0, service = 1) was entered as the predictor (X), dummy coded award type (teaching, and impact), dummy coded award level, shared, honorable mentions, as well as interaction terms for these variables were added as covariates to the model. In the second model, the teaching dummy (research = 0; teaching = 1) was entered as the predictor (X), and other variables were the same, except that service dummy and its interaction with year were also added as covariates.

### Reporting summary

Further information on research design is available in the Nature Portfolio Reporting Summary linked to this article.

## Results

First, we report descriptive statistics, followed by some tests of gender disparities across categories. Next, we report our focal analyses of change over time in social and personality award giving. For the sake of simplicity, in our analyses we collapse the early career researcher (ECR), postgraduate (PG), and PG/ECR categories into a single category (although note that this does not substantially change any conclusions).

### Awards in social and personality psychology

A total of 2700 awards were given between 1968 and 2021. As shown in Table 1, most awards were given to men, followed by women, and individuals for whom gender/sex could not be ascertained or was given to an entity rather than an individual. We examined whether gender disparities in awards were present when compared to a 50:50 benchmark. While crude, we chose this benchmark to avoid difficulties of ascertaining accurate base rates of social and personality psychologists internationally. Meaningful base rates are often only available for the US (e.g., PhD graduates) or for single societies (e.g., EASP). That being said, from available society specific data, 50:50 can be considered a conservative test as women are the majority members (for example, 54% and 54.1% for SPSP and EASP, respectively). Men received awards above expected frequencies (observed $n = 1571$) and women received awards below expected frequencies (observed $n = 1121$), $\chi^2(1) = 75.22$, $p < 0.001$, $\omega = 0.17$.

Next, we examined the features of awards by gender/sex recipient. Men received more research (60.9%, $n = 1295$), and impact/media (68.7%, $n = 92$) awards than expected, whereas women received more service (49.8%, $n = 123$), and teaching awards (67.4%, $n = 124$) than expected, $\chi^2(3) = 68.39$, $p < 0.001$, $\varphi_c = 0.16$, 95% CI [0.12, 0.20]. Sub-sample analyses comparing likelihood of women receiving awards at each level individually compared to men were conducted. For research ($\chi^2(1) = 26.11$, $p < 0.001$, OR = 0.61, 95% CI [0.51, 0.74]) and impact ($\chi^2(1) = 5.72$, $p = 0.017$, OR = 0.63, 95% CI [0.43, 0.91]) the likelihood of a woman winning was 0.61 and 0.63 as high as the likelihood of a man winning these award types. For teaching ($\chi^2(1) = 52.75$, $p < .001$, OR = 3.13, 95% CI [2.28, 4.31]) and service ($\chi^2(1) = 7.08$, $p = .008$, OR = 1.44, 95% CI [1.11, 1.87]) awards women were 3.13 and 1.44 times more likely to win than men.

We also observed significant differences for award level overall, $\chi^2(3) = 35.39$, $p < 0.001$, $\varphi_c = 0.12$, 95% CI [0.08, 0.15]. Descriptives suggested men received more awards at senior (65.8%, $n = 505$), and mid-career levels (69.6%, $n = 48$) than expected, but not at PG/ECR and all levels. At PG/ECR and all levels, women took home more awards than expected (47.9%, $n = 348$, and 44.1%, $n = 472$, respectively). Further sub-sample analyses showed that women were 0.63 times as likely to win senior awards ($\chi^2(1) = 26.14$, $p < 0.001$, OR = 0.63, 95% CI [0.53, 0.75]); and 1.40 times more likely to win at PG/ECR level ($\chi^2(1) = 14.40$, $p < 0.001$, OR = 1.40, 95% [1.18, 1.66]). While descriptively women and men won more awards at all levels and mid-career respectively sub-sample analyses did not show significant differences ($\chi^2(1) = 3.50$, $p = 0.061$, OR = 1.17, 95% CI [1.00, 1.36], and $\chi^2(1) = 3.35$, $p = 0.067$, OR = 0.60, 95% CI [0.36, 1.01]).

We also examined the breakdown of awards to women and men by award type and level simultaneously. As shown in Fig. 1, men received most awards at each level for research and impact/media awards. For teaching, the reverse was observed – women received most awards at each level. For service, the picture was more balanced with women and men dominating at different levels.

For shared awards, women received shared awards (43.5%, $n_{Women} = 767$) at a higher rate than expected, and fewer individual awards than expected (38.1%, $n_{Women} = 354$). For men, the opposite was true (56.5%, $n_{Shared} = 997$; 61.9%, $n_{Not-shared} = 574$). The Pearson's Chi Square test was significant, $\chi^2(1) = 6.90$, $p = 0.009$, OR = 1.25, 95% CI [1.06, 1.47] indicating that women were 1.25 times more likely to win a shared award than men. Finally, 49.5% of honorable mentions were for women ($n = 138$), and 50.5% for men ($n = 141$). A Pearson's Chi Square test was significant, $\chi^2(1) = 7.48$, $p = .006$, OR = 1.42, 95% CI [1.11, 1.83] indicating that women were 1.42 more likely to receive an honorable mention than men were.

### Change in social and personality psychology awards over time

We also examined trends over time. As shown in Fig. 2, women increased their share of awards over time, but men have continued to receive the majority of awards. Albeit, in the most recent years (2020–2021) women have outstripped men in awards received (i.e., 75 vs 43 and 49 vs. 41, respectively). Further, Fig. 3 shows a breakdown of the type and level of the awards won by women and men overtime. As shown in panels a and b, most awards won by women and men over time are for research. For women (panel b) teaching and service awards appear to be the next most frequent type of awards won, whereas for men (panel a) the number of awards given for service, impact, and teaching appear more balanced over time. As shown in the bottom panels, most awards won by women (panel d) were at PG/ECR or all levels over time, with a lower number of senior awards won over time. In contrast for men (panel c), the number of PG/ECR, all level and senior awards was more equal over time. The same pattern was found when we examined proportions of awards given over time. Please see Supplementary Figs. 2–5 for full details.

Next, we conducted a logistic regression to investigate the joint associations between the different award features and gender/sex of the winner. The model was significant $\chi^2(9) = 158.65$, $p < .001$ and explained 7.9% (Nagelkerke $R^2$) of the variance. As shown in Table 3, year, award type and

**Fig. 1 | Proportion of awards given to women and men split by award type and level.** All available data included except winners categorized as not applicable for gender/sex. PG Postgraduate; ECR Early Career Researcher, NA Not Applicable. Green indicates women; yellow indicates men. **a** Research awards, **b** Impact and media awards, **c** Service awards, **d** Teaching awards.

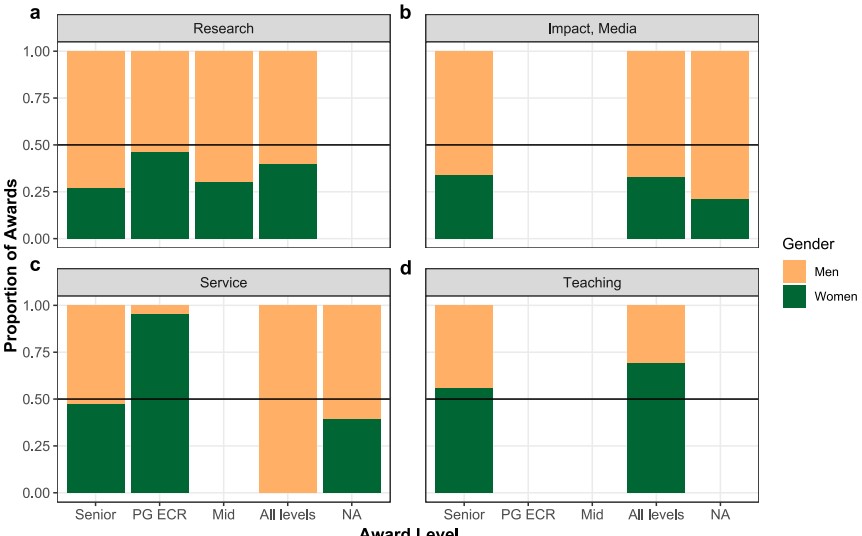

award level were significant predictors of recipient gender/sex. The winner was more likely to be a woman if the award was later in time, was for service, or teaching, compared to research contributions, and if the award was PG/ECR, or all levels compared to senior. No other category comparisons or predictors were significant.

Next, we conducted a logistic regression to investigate whether award features (type, level, shared and honorable mentions) were moderated by year. The overall model was significant $\chi^2(17) = 178.09$, $p < 0.001$ and explained 8.8% (Nagelkerke $R^2$) of the variance. As shown in Fig. 4, significant interactions were present for award type, specifically year x service (vs. research), $b = 0.04$, $SE = 0.02$, $p = 0.045$, OR = 1.04, 95% CI [1.00, 1.09], and year x teaching (vs. research), $b = 0.08$, $SE = 0.03$, $p = 0.018$, OR = 1.08, 95% CI [1.01, 1.15].

To probe these interactions further, we reran this model with the Johnson-Neyman technique using the Process macro for SPSS (Model 1[44]). The first model tested the influence of the service dummy (research = 0, service = 1). After 1999, winners of service (vs. research) awards were significantly more likely to be women than men. The second model tested the teaching dummy (research = 0, teaching = 1). For awards after 2007, winners of teaching (vs. research) awards were significantly more likely to be women than men. See Supplementary Figs. 6 and 7 for plots of these analyses. All other interactions were non-significant ($p$s > 0.140). Note, when we analyzed interactions for each predictor separately the year x all levels (vs. senior) interaction was significant ($p = 0.017$, OR = 0.97, 95% CI = [0.95, 1.00]).

## Discussion

We investigated gender disparities in social and personality psychology awards from 17 societies over five decades. We examined whether the proportion of awards given to women (vs. men) has changed over time, and whether it depends on the type of award (e.g., research, service) and the level of the award (e.g., early career, senior). Overall, men were more likely to win awards than women, but women were more likely to win awards for service or teaching compared to research, and more likely to win junior or all level (vs. senior) awards. These findings were consistent with our predictions. We further found that women were more likely to win awards over time, although this trend seems to be partly explained by the fact that service and teaching are more likely to be recognized with awards in recent years. Specifically, awards for service and teaching were more likely to be given to women, compared to men, after 1999 and 2007 respectively.

Overall, our findings align with previous research demonstrating gender gaps in awards[35,39]. In the present study focusing on social and personality psychology, women took home 41.5% of awards overall, which

is considerably larger than estimates in other disciplines (since 2000) including physics (8.8%), chemistry (10.3%), and biomedicine (18.5%[45]). Comparisons within the discipline of psychology show a similar pattern – women took home a greater number of awards in our sample than APA recognition award winners across an equivalent timeframe (i.e., 27.4% women[40]). Together, these findings depict a relatively positive picture of social and personality psychology award giving. However, direct comparisons are limited by the heterogeneity across investigations. Unlike social and personality psychology prizes, those in natural sciences are often more prestigious with financial rewards attached.

Our findings are also consistent with social psychological theorizing on gender stereotyping and roles. We found that women were more likely to win awards for teaching and service rather than research contributions, which may reflect implicit stereotypes of women as more communal and concerned with helping others[46]. Further, when it comes to the most senior and prestigious awards, we found that women were less likely to win in these categories and more likely to win at junior or all levels. These findings may reflect culturally prevalent beliefs suggesting women lack sufficient brilliance to meet thresholds for senior or prestigious contributions[26]. Moreover, they are consistent with recent research showing women receive less prestigious awards than men[34] and fewer invited submissions to prestigious outlets[47], which might give some indication of how women's pursuit of prestige is frustrated earlier in the pipeline.

Turning to the effects of time on award giving in social and personality psychology, we found that over the years women have increased their share of awards. In 2020-2021, women even outstripped men in total awards. Such a stark change in pattern suggests the potential influence of acute cultural events. The #MeToo movement was a watershed moment for gender relations in 2017-18, which spotlighted the persistence of gender inequality in many sectors[48]. While our data cannot test this, it is possible that greater collective awareness of gender bias influenced award giving in the following years. For example, there is evidence that in the wake of #MeToo, the Nobel committee has explicitly asked scientists to consider gender/sex when making nominations[49]. We also observed an overall drop in awards given in 2021. This may reflect the cancellation or postponement of conferences and society meetings due to the Covid-19 pandemic[50]. Further research should continue to track gender/sex disparities in award giving to disentangle the influence of one-off events from longer term factors.

Alternatively, our data hint at the possible influence of a long-term trend by showing that the proliferation of awards to women is in part driven by the tendency for women to win certain awards – namely for service and teaching contributions. This finding is consistent with recent research from biomedicine showing increases in women prize winners over a similar

timeframe from 5% to 27%, but like psychology, this is explained by greater recognition of women's contributions to service, teaching and mentoring[4]. It is also worth considering our findings in light of recent evidence showing changes in stereotyping. Research shows implicit stereotyping of men with science/career and women with arts/family has decreased between 2012 and 2018; albeit the overall tendency towards stereotypical associations remains[19]. Relatedly, analysis of polling data from 1946 to 2018 shows that while stereotypes of women as incompetent have decreased (from 66% to 35%) stereotypes regarding women's greater communion have increased (from 54% to 97%[18]). Our findings could reflect this shift: women are now

seen as competent enough to be scientific winners, but more likely to win in categories which are congruent with communality.

These findings – in particular, the dearth of women winners in the most senior and prestigious award categories – have important implications for the field of psychology. While the overall disparity in awards won may not seem large, awards are an important indicator of eminence within a field and help shape who is viewed as an important and influential thinker[10,35]. Ma and Uzzi[51] found that, despite an increase in the number of prizes overall, a small number of highly clustered scientific elites win a large portion of these prizes. These scientists tend to be tightly intertwined in co-authorship networks, which can lead to in-group biases and work against the ethic of inclusiveness within science. This clustering of prizes can also grant legitimacy to certain ideas and steer the course for which knowledge pathways are formed between different sub-disciplines of psychology, or even between psychology and separate disciplines altogether. For example, prolific psychologist Daniel Kahneman claimed that winning the Nobel Prize in Economics in 2002 legitimated his psychological research findings and led to an increase in the knowledge transfer between psychology and economics[52]. As such, gender/sex disparities in the presentation of awards can have important downstream effects on other aspects of academic success, including whose ideas are allowed to become influential both within the field and outside of it. Identifying the existence of disparities is the first step towards potentially rectifying it in the future.

## Limitations

Further research is needed to explain the reasons why women are underrepresented in some types of social and personality psychology awards. Research from neuroscience points to the influence of total citations and *h*-indices as important predictors in prestigiousness of the awards women win[34]. Such metrics may explain why women are winning less research awards, but the extent to which they influence other types of awards is not

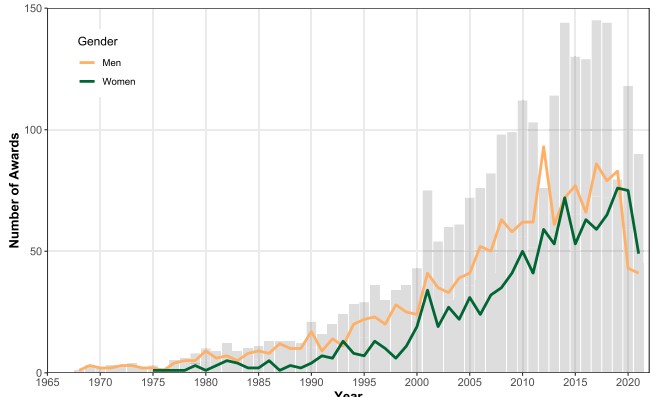

**Fig. 2 | Numbers of awards in social and personality psychology from 1968 to 2021 in total, among women, and among men.** Grey bars represent the total number of awards given a year. Lines represent the number of awards given to women (in green) and men (in yellow) in a given year. All available data included.

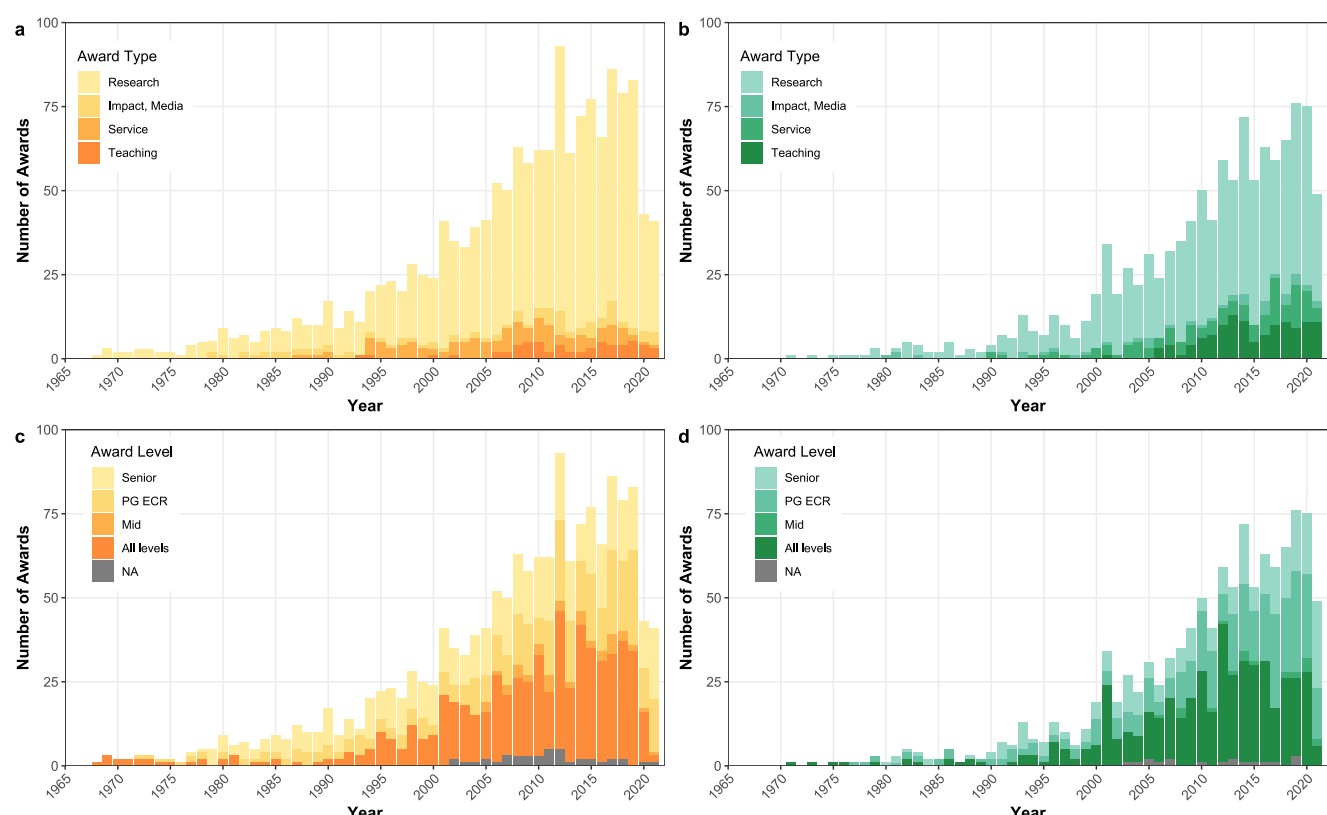

**Fig. 3 | Numbers of awards in social and personality psychology from 1968 to 2021 among women and men separately, and split by award type and level.** PG postgraduate, ECR Early career researcher, NA not applicable. Green indicates

women; yellow indicates men. **a** Men award winners by award type, **b** Women award winners by award type; **c** Men award winners by award level, **d** women award winners by award level.

**Table 3 | Logistic regression analysis for variables predicting women (vs. men) award recipients**

| Predictor | b | SE | Wald | OR | 95% CI OR Lower, Upper | | p |
|---|---|---|---|---|---|---|---|
| Year | 0.03 | 0.00 | 34.52 | 1.03 | 1.02 | 1.04 | <0.001 |
| Award Type: reference Research | | | 62.41 | | | | <0.001 |
| Impact/Media | 0.37 | 0.23 | 2.56 | 1.45 | 0.92 | 2.28 | 0.110 |
| Service | 0.93 | 0.17 | 29.94 | 2.54 | 1.82 | 3.55 | <0.001 |
| Teaching | 1.03 | 0.18 | 34.35 | 2.80 | 1.99 | 3.95 | <0.001 |
| Award Level: reference Senior | | | 41.85 | | | | <0.001 |
| Mid-Career | 0.00 | 0.28 | 0.00 | 1.00 | 0.58 | 1.75 | 0.988 |
| PG/ECR | 0.79 | 0.13 | 38.50 | 2.21 | 1.72 | 2.84 | <0.001 |
| All levels | 0.52 | 0.13 | 14.91 | 1.68 | 1.29 | 2.18 | <0.001 |
| Shared | −0.01 | 0.11 | 0.00 | 0.99 | 0.81 | 1.22 | 0.948 |
| Hon. Mention | 0.24 | 0.15 | 2.81 | 1.28 | 0.96 | 1.69 | 0.094 |

*N* = 2629. *Hon. Mention* Honorable Mention, *PG* Postgraduate, *ECR* Early career researcher.

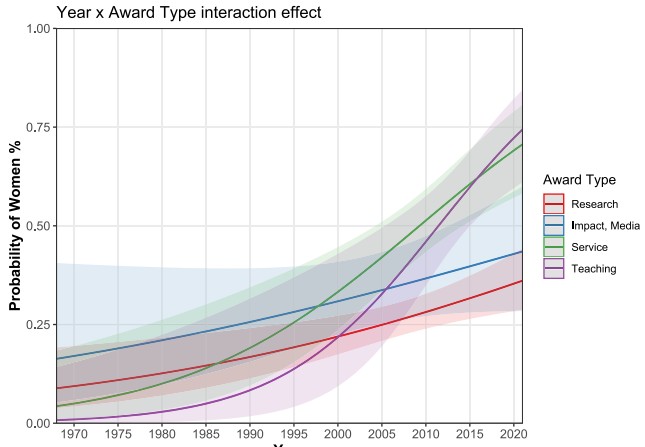

**Fig. 4 | Probability of award recipient being a woman by award type over time.** Line represents the slope, and colored area represents the 95% confidence intervals. Red indicates research; blue indicates impact and media; green indicates service; purple indicates teaching.

clear. Others have hinted at the influence of selection panels and practices as important mechanisms through which women may be disadvantaged. Lincoln and colleagues[6] report women's odds of winning are increased when more women are on the selection committee – however, if a man is committee chair, any effect women might have is lost. In fact, in their data when men chaired selection committees, they awarded prizes to men 95.1% of the time, despite women making up 21% of the overall pool of nominees being considered by these committees. These findings might be partially explained by recent work on biases in memory accessibility[14]. A survey of R1 US psychology faculty showed that faculty who were men were more likely to recall experts in the field who were men than those who were women[14]. This bias might not only help explain discrepancies in who is nominated for awards, but also who is cited, invited, or nominated for important roles. Beyond procedural changes, wider impacts of gender roles on awards disparities need to be examined further. Perhaps most important are the impacts of childrearing on academic careers – women are more likely than men to take time off for childcare, and can often disappear from publishing while their children are young in a way that men rarely do[18]. Lincoln and colleagues[6] stress that these gender norms also influence how individual women assess their own ability; thus, women self-promote and seek award nominations at a lower rate than men of equal ability. This is further exacerbated by award criteria that relies heavily on masculine-coded language, such as calling for leaders or risk-takers.

A potential limitation of the current investigation is the way in which gender/sex of award winner was categorized by relying on the extent to which individual appearance conforms to gendered expectations (i.e., normative gendered presentation). While care was taken to assess different sources to ascertain gender/sex (including not only photos and given names, but also pronoun use in CVs, personal and professional web pages), these metrics cannot account for winner's self-identification. This is particularly pertinent for those who identify outside of the gender/sex binary, or whose gender presentation does not reflect their felt identity. Likewise, other biases in award giving related to sexuality were not addressed in our investigation and present important avenues for future research.

Finally, further research should examine racial and ethnic biases in awards. While one might assume this is also a widespread problem within psychology and academia more broadly, comprehensive studies on this topic are largely lacking[53]. One of the few papers on race, ethnicity and awards found that African American applicants were 10 percentage points less likely than their white colleagues to receive U.S. National Institutes of Health research funding, even after controlling for variables such as educational background, training, and publication record[54]. Another issue related to these biases is the potential Anglophone bias within psychology. Because the English language has such global dominance and is used so widely within media more broadly as well as scientific literature, a hierarchy of knowledge is created whereby non-Anglophone scholars are at a disadvantage within scientific research networks compared to native English-speakers[55]. The impact of Anglophone-dominance within psychology award recipients is another area in need of investigation.

## Conclusions
While the widespread nature of gender/sex disparities in award distribution throughout academia can make achieving gender/sex equity seem like a momentous task, the present findings suggest that the tide may be changing in the recognition of women's contributions to psychological science. Despite these improvements, efforts need to be made to make sure women's contributions are recognized equally across career stages, type of activity, and prestige. Psychology in general – and social and personality psychology in particular – is uniquely suited to handling problems of this nature. Given the field's vested interest in understanding the role of gender stereotyping and other biases in human behavior and decision-making, psychological scientists can work to take gender disparities into account when selecting award recipients and potentially serve as a catalyst for other fields to do the same[37]. Gender/sex disparities in psychological awards is a pressing issue with real-world consequences for women in the field[56], and working towards recognizing the issue can help us take steps toward changes that will ensure a more equal playing field for psychologists in the future.

## Data availability

The data that support the findings of this study are openly available at the Open Science Framework (OSF) at https://osf.io/8dbyr/.

## Code availability

The custom analysis code is also openly available at the Open Science Framework (OSF) at https://osf.io/8dbyr/[57]. The data were analyzed using SPSS version 28/29, Process macro for SPSS version 4.2, and R version 4.3 packages tidyverse, sjplot, lme4, lmerTest, cowplot, interactions, sandwich.

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

## Acknowledgements
We would like to thank the Editor and reviewers for their thoughtful feedback on this research. Thanks also to members of the Political Psychology Lab at the University of Kent and the Gender and Sexuality Lab at University of Surrey for their feedback on earlier drafts.

## Author contributions
A.H.-D. and J.C. contributed equally. Conceptualization: A.C., A.H.-D.; Data Curation: J.C, A.H.-D.; Formal Analysis: A.H.-D., D.T.F, A.C.; Funding Acquisition: N.A.; Investigation: J.C.; Methodology/Resources: A.H.-D., J.C., A.C.; Project Administration: A.H.-D., J.C.; Validation: A.H.-D., D.T.F.; Visualizations: D.T.F., J.C.; Writing - original draft: A.H.-D., J.C.; Writing - review and editing: A.H.-D., J.C., A.C., D.T.F.

## Competing interests
The authors declare no competing interests.
