## [Peer Review File · Communications Psychology]

8th Dec 23

Dear Aífe,

Thank you for your patience during the peer-review process. Your manuscript titled "Gender Disparities in Professional Social and Personality Psychology Awards from 1968 to 2021" has now been seen by 2 reviewers, and I include their comments at the end of this message. They find your work of interest, but raised some important points. We are interested in the possibility of publishing your study in Communications Psychology, but would like to consider your responses to these concerns and assess a revised manuscript before we make a final decision on publication.

We therefore invite you to revise and resubmit your manuscript, along with a point-by-point response to the reviewers. Please highlight all changes in the manuscript text file.

Editorially, we consider it important that you address the requests for further analysis made by Reviewer #1. As for the request to motivate the focus on psychology, as we are a journal dedicated to the (broad) psychology community, this would not be strictly necessary. However, the Discussion may contain a succinct review of similarities and differences between fields (as evident from the literature). Reviewer #2 invites you to reflect more on the reasons for recent changes and integrate existing literature. Some speculation is permitted in the Discussion but should be suitably caveated in the Limitations section.

Please note that your revised manuscript must comply with our formatting and reporting requirements, which are summarized on the following checklist:

Communications Psychology formatting checklist and also in our style and formatting guide Communications Psychology formatting guide .

Please use the following link to submit your revised manuscript, point-by-point response to the referees' comments (which should be in a separate document to any cover letter) and the completed checklist:

[link redacted]

Please do not hesitate to contact me if you have any questions or would like to discuss these revisions further. We look forward to seeing the revised manuscript and thank you for the opportunity to review your work.

Best regards,

Marike

Marike Schiffer, PhD

Chief Editor

Communications Psychology

EDITORIAL POLICIES AND FORMATTING

Editorial Policy: Policy requirements (Download the link to your computer as a PDF.)

* **CODE AVAILABILITY:** All Communications Psychology manuscripts must include a section titled "Code Availability" at the end of the methods section. In the event of publication, we require that the custom analysis code supporting your conclusions is made available in a publicly accessible repository; at publication, we ask you to choose a repository that provides a DOI for the code; the link to the repository and the DOI will need to be included in the Code Availability statement. Publication as Supplementary Information will not suffice. We ask you to prepare code at this stage, to avoid delays later on in the process.

* **DATA AVAILABILITY:**

All Communications Psychology manuscripts must include a section titled "Data Availability" at the end of the Methods section or main text (if no Methods). More information on this policy, is available at <http://www.nature.com/authors/policies/data/data-availability-statements-data-citations.pdf>.

At a minimum the Data availability statement must explain how the data can be obtained and whether there are any restrictions on data sharing. Communications Psychology strongly endorses open sharing of data. If you do make your data openly available, please include in the statement:

We recommend submitting the data to discipline-specific, community-recognized repositories, where possible and a list of recommended repositories is provided at <http://www.nature.com/sdata/policies/repositories>.

If a community resource is unavailable, data can be submitted to generalist repositories such as figshare or Dryad Digital Repository. Please provide a unique identifier for the data (for example a DOI or a permanent URL) in the data availability statement, if possible. If the repository does not provide identifiers, we encourage authors to supply the search terms that will return the data. For data that have been obtained from publicly available sources, please provide a URL and the specific data product name in the data availability statement. Data with a DOI should be further cited in the methods reference section.

REVIEWERS' EXPERTISE:

Reviewer #1 metascience, social science

Reviewer #2 metascience, psychology

REVIEWERS' COMMENTS:

Reviewer #1 (Remarks to the Author):

Review of Gender disparities in Psychological awards, Nature Communications.

This paper investigates patterns of gender equality in scholarly awards in the discipline of psychology. The data is a relatively large sample of 17 prizes won by 2701 prizewinner 1968 and 2021. Care was taken by the authors to identify prize characteristics and the gender of the winner. Regression analyses confirm raw data findings that men compared to women are overrepresented as prize winners of research awards but that the opposite is true of teaching and service. Recently women have been closing the gap in research awards but the inequality is still significant.

I think the paper can be a strong contribution to psychology and should be published after the authors fine-tune their review of the literature and regression model.

Introduction:

I think the authors should update their introduction to be more aligned with the current literature and to make it relevant to a general science audience. The current literature emphasizes the broad impact that prizes on a scientific field. For example, Harriet Zuckerman's foundational work (1970, 1972) showed that prizes uniquely motivate greatness, incentive high quality, create role models and (gender) stratification in science. Other work shows that prizes lead ideas in science and scientists strongly investment in topics that win prizes (Ma and Uzzi 2018; Jin et al 2021) and that prizes provide special mentorship and create stratification in science (Ma et al. 2020).

Related to updating the introduction, which needs to attend to the interest of non-psychology readers, the opposite is also true. The authors should say more about why the study of gender equality and prizes in psychology is particularly interesting and set apart from prior work. This is done to some degree by talking about the high participation rate of women in psychology relative to other fields, the fact that psychology graduate programs now train more women than men, and so on. Nevertheless, I felt the psychology specific reasons for studying psychology should be drawn together into a more cogent argument. Another reason to strengthen the motivation around studying just psychology is that other work in biology and physics shows that women are increasing in representation of prizes but are still under-represented in research prizes and over-represented in service prizes (Ma et al 2019, Jin and Uzzi2021) would focus the readers on new knowledge for psychology and science.

Regression analysis needs further clarification.

It is not clear whether Year was entered as a dummy variable, i.e., a fixed effect for each year or a trend. If you add year to the regression as a scalar variable, the variable simple shows the correlation with time advancing the probably of a male vs female prize recipient. To control for unobserved variables that do not vary within year such as number of scientists in psychology, number of journals, number of prizes, number of graduates, the proportion of women and men scholars and so on , each Year must be added as a fixed effect variable or several years should be combined into separate period effects. In this way, you would get a separate intercept estimated by the regression for each year.

A related question is that your sample of prizes tend to be concentrated in a few prizes that make up 10% or more of the prizes. I would be keen to know what happens to the results when you add fixed effects for these prizes in the regressions since your results are being driven by the prizes with the greatest contribution to the sample and these prizes may not be generalizable to the rest of psychology prizes. For example, different societies may have particular nomination processes, offer money, be especially prestigious, or emphasize certain psychology subfields and so on.

I would also be keen to see what happens to the regression if you add a scholars career length (time since first publication) to the regression.

For the proportion of awards figures, does the 50% horizontal line make sense? Perhaps a line that shows the participation rate of women for each year could be added because it is relevant when the participation rate is less than or more than 50%

The figures, plots and tables need figure captions.

Reviewer #2 (Remarks to the Author):

This is an important, well-executed, and clearly written paper that I endorse for publication. I have only a few comments, which I hope the authors will use to further refine their already excellent manuscript.

My main question pertains to the recent reversal in the gender disparity in awards—in particular, what its underlying cause(s) might be. The authors speculate that the shift can be explained in terms of changing stereotypes, with women being viewed as more competent and communal over time. This may well be a contributing factor, but it seems like a relatively small factor at best, because the shift towards parity in awards was highly nonlinear; there was a sudden, massive shift starting in 2020 that seems unlikely to (fully) reflect a gradual shift in gender stereotypes. I think it would be worth speculating about what could have happened in recent years to cause such a drastic shift in award distribution. This is part because different underlying causes have different implications for what we can expect the future to hold. If the shift were due entirely to shifts in stereotyping, we could expect the current, relatively equitable state of affairs to persist (or even grow) in the years ahead. But if the shift we saw in 2020 was due to some acute cultural event, we

might expect that progress to dissipate as time goes on. So, I think the nonlinearity of the recent shift towards equity merits further discussion.

A couple minor comments:

On lines 54-55, the authors distinguish descriptive from prescriptive elements of gender stereotypes. However, it's not clear what role this conceptual distinction plays in the authors' theorizing, or how it bears on the results. In fact, as far as I can tell, the descriptive/prescriptive distinction is not mentioned at all outside that one sentence. I suggest the authors either drop that sentence or clarify how it is relevant.

The authors discuss work showing that explicit stereotyping of women has changed over the decades, but there is also work by Charlesworth and colleagues (e.g., Charlesworth et al., 2022, Patterns of Implicit and Explicit Stereotypes III: Long-Term Change in Gender Stereotypes) showing a similar pattern at the implicit level. I think that citing and discussing this work would add value. In addition, the authors might consider mentioning in the introduction that gender stereotypes have become more egalitarian over time, as this would help set the stage for the results that follow.

Communications Psychology Response to Reviewers' Comments

Manuscript: "Gender Disparities in Professional Social and Personality Psychology Awards from 1968 to 2021"

REVIEWERS' COMMENTS:

Reviewer #1 (Remarks to the Author):

Review of Gender disparities in Psychological awards, Nature Communications.

This paper investigates patterns of gender equality in scholarly awards in the discipline of psychology. The data is a relatively large sample of 17 prizes won by 2701 prizewinner 1968 and 2021. Care was taken by the authors to identify prize characteristics and the gender of the winner. Regression analyses confirm raw data findings that men compared to women are overrepresented as prize winners of research awards but that the opposite is true of teaching and service. Recently women have been closing the gap in research awards but the inequality is still significant.

I think the paper can be a strong contribution to psychology and should be published after the authors fine-tune their review of the literature and regression model.

Thank you for your thorough review of our work.

Introduction:

I think the authors should update their introduction to be more aligned with the current literature and to make it relevant to a general science audience. The current literature emphasizes the broad impact that prizes on a scientific field. For example, Harriet Zuckerman's foundational work (1970, 1972) showed that prizes uniquely motivate greatness, incentive high quality, create role models and (gender) stratification in science. Other work shows that prizes lead ideas in science and scientists strongly investment in topics that win prizes (Ma and Uzzi 2018; Jin et al 2021) and that prizes provide special mentorship and create stratification in science (Ma et al. 2020)."

Thank you for this suggestion. We have decided not to broaden the scope of the introduction to a more general science audience given that Communications Psychology is a discipline specific outlet (as the Editor states), concerns about the length of the introduction, and because we already discuss the findings of Ma et al (2019) on p 4 and Ma and Uzzi (2018) on p 22. However, following your helpful suggestion we have added the work of Jin et al. (2021) and Zuckerman (1970) on p 1, and later on p 22.

Related to updating the introduction, which needs to attend to the interest of non-psychology readers, the opposite is also true. The authors should say more about why the study of gender equality and prizes in psychology is particularly interesting and set apart from prior work. This is done to some degree by talking about the high participation rate of women in psychology relative to other fields, the fact that psychology graduate programs now train more women than men, and so on. Nevertheless, I felt the psychology specific reasons for studying psychology should be drawn together into a more cogent argument. Another reason to strengthen the motivation around studying just psychology is that other work in biology and physics shows that women are increasing in representation of prizes but are still under-represented in research prizes and over-represented in

service prizes (Ma et al 2019, Jin and Uzzi2021) would focus the readers on new knowledge for psychology and science.

Thank you for this suggestion. In line with the Editor's suggestions, we have not sought to motivate the focus on psychology further given that Communications Psychology is a discipline specific outlet. However, following your and the Editor's suggestion, we have added a few sentences throughout the discussion section comparing our findings with the wider literature from other disciplines to focus the readers on new knowledge from psychology and science. Now on page 20-22:

p 20-21

“Overall, our findings align with previous research demonstrating gender gaps in awards^{35,39}. In the present study focusing on social and personality psychology, women took home 41.5% of awards overall, which is considerably larger than estimates in other disciplines (since 2000) including physics (8.8%), chemistry (10.3%), and biomedicine (18.5%⁴⁶). Comparisons within the discipline of psychology show a similar pattern – women took home a greater number of awards in our sample than APA recognition award winners across an equivalent timeframe (i.e., 27.40% women⁴⁰). Together, these findings depict a relatively positive picture of social and personality psychology award giving. However, direct comparisons are limited by the heterogeneity across investigations. Unlike social and personality psychology prizes, those in natural sciences are often more prestigious with financial rewards attached.”

And p 22:

“Alternatively, our data hint at the possible influence of a long-term trend by showing that the proliferation of awards to women is in part driven by the tendency for women to win certain awards – namely for service and teaching contributions. This finding is consistent with recent research from biomedicine showing increases in women prize winners over a similar timeframe from 5% to 27%, but like psychology, this is explained by greater recognition of women's contributions to service, teaching and mentoring⁴.”

On p 24, we have also included some new research from psychology which offers new ways of interpreting existing research on awards from science more broadly:

“These findings might be partially explained by recent work on biases in memory accessibility¹⁴. A survey of R1 US psychology faculty showed that male faculty were more likely to recall male than female experts in the field¹⁴. This bias might not only help explain discrepancies in who is nominated for awards, but also who is cited, invited, or nominated for important roles.”

Regression analysis needs further clarification. It is not clear whether Year was entered as a dummy variable, i.e., a fixed effect for each year or a trend. If you add year to the regression as a scalar variable, the variable simple shows the correlation with time advancing the probably of a male vs female prize recipient. To control for unobserved variables that do not vary within year such as number of scientists in psychology, number of journals, number of prizes, number of graduates, the proportion of women and men scholars and so on , each Year must be added as a fixed effect variable or several years should be combined into separate period effects. In this way, you would get a separate intercept estimated by the regression for each year.

Thank you for this suggestion. The model reported in our original manuscript entered Year as a continuous variable. This model was used to test basic associations between variables before

examining interactions between Year and Award Level/Type which was of greater interest to us. We investigate these interactions in a follow-up regression model and using the Johnson-Newman technique to pinpoint the exact year when the Year x Award Level/Type effect became significant. These are now shown on p 19-20. Below we report the findings of the additional analysis you suggest, however given the only minor differences across models we have chosen not to report this in the main text.

Following your suggestion, we have re-run the regression model entering fixed effects for time at 5-year intervals (dummy variables). Doing this did not substantially change the findings. The overall model was significant, $\chi^2(17) = 163.86$, $p < .001$, and explained 8.1% of the variance (a 0.2% increase). Consistent with our original model, the award was more likely to be given to a woman when given for Teaching or Service (vs Research) and when give at PG/ECR or All levels (vs Senior). For Year, the award was more likely to be given to a woman in the year 2001-05 (OR = 4.74, 95% CI [1.28, 15.60], $p = .019$), 2006-10 (OR = 3.87, 95% CI [1.12, 13.42], $p = .033$), 2011-15 (OR = 4.56, 95% CI [1.32, 15.75], $p = .017$), and 2016-21 (OR = 5.62, 95% CI [1.63, 19.41], $p = .006$) compared to earlier years (1968-1975 was the reference category).

A related question is that your sample of prizes tend to be concentrated in a few prizes that make up 10% or more of the prizes. I would be keen to know what happens to the results when you add fixed effects for these prizes in the regressions since your results are being driven by the prizes with the greatest contribution to the sample and these prizes may not be generalizable to the rest of psy society prizes. For example, different societies may have particular nomination processes, offer money, be especially prestigious, or emphasis certain psychology subfields and so on.

Thank you for this suggestion. We have identified one prize which comprised > 10% of total prizes given (i.e., Gordon Allport Intergroup Relations prize from SPISSI, 11.8%) and created a fixed effect for this prize (coded 1, all other categories coded 0). We then re-ran the model using this dummy. These additions do not substantially change the findings, which produced a small increase in variance explained of 0.5% (Overall model: $\chi^2(10) = 169.05$, $p < .001$, $R^2 = .084$) and the same pattern was present for Year, Award Type and Award Level. The effect of the Gordon Allport prize was significant, indicating that the award winner was more likely to be female when the award was the Gordon Allport prize (vs all other prizes) $b = .48$, $SE = 0.15$, $Wald = 10.43$, $OR = 1.61$, 95% CI [1.21, 2.16], $p = .001$. The next two highest proportions of prizes (i.e., > 5%) were given for the Otto Klineberg Intercultural and International Relations (5.3%; SPISSI) and the "Dissertation" SESP award (5.2%, SESP). We also created dummies for these and added them to the model with Gordon Allport. The same pattern was present for Year, Award Type, Award Level, and Gordon Allport, and in addition the Otto Klineberg award explained 0.02% of variance. The award winner was more likely to be a woman when the award was Otto Klineberg ($b = .43$, $SE = .20$, $Wald = 4.87$, $OR = 1.54$, 95%CI [1.05, 2.27], $p = .027$). No significant effect was found for the SESP Dissertation award ($p = .508$). Note that both the Gordon Allport and Otto Klineberg awards are given by SPISSI which is also the society which has awarded the majority of awards between 1968 and 2021 at 30.5% of awards. These findings suggest that two SPISSI awards appear to contribute to the overall effect but not over and above our predicted effects of Year, Award Level, and Award Type.

I would also be keen to see what happens to the regression is you add a scholars career length (time since first publication) to the regression.

This is a great suggestion but unfortunately, we are not able to address this given the complexities of tracking authors internationally and across such a long time frame (i.e., 5 decades). Academic conventions in respect of documenting publications have changed significantly over time. For

example, there may be no digital record (e.g. google scholar, ORCID) of first publication for some academics who won prizes earlier in time. Likewise, given the time that has elapsed we were not even able to identify some recipients and instead used the gender API tool to identify gender from their first name (<https://gender-api.com/en/>). Further, our current analyses already account to some extent for such differences by categorizing awards by career stage (i.e., ECR vs Senior career stages).

For the proportion of awards figures, does the 50% horizontal line make sense? Perhaps a line that shows the participation rate of women for each year could be added because it is relevant when the participation rate is less than or more than 50%

We agree that a line showing participation rates of women in a given year would be a meaningful benchmark in these figures. However, it is not possible to get accurate estimates of rates of women in a given year because of the international nature of the data. As we outline on p 12-13, such data is only available in the US, however, it only reflects the number of PhDs awarded to women in that given year:

“We examined whether gender disparities in awards were present when compared to a 50:50 benchmark. While crude, we chose this benchmark to avoid difficulties of ascertaining accurate base rates of social and personality psychologists internationally. Meaningful base rates are often only available for the US (e.g., PhD graduates) or for single societies (e.g., EASP). That being said, from available society specific data, 50:50 can be considered a conservative test as women are the majority members (for example, 54% and 54.1% for SPSP and EASP, respectively).”

There are also problems with this operationalization as it does not accurately reflect participation rates of women eligible for awards in a given year. On further reflection, and in interest of saving space and writing a succinct results section, we have decided to summarize the proportional findings in text and move these figures (i.e., Figures 4-7 in the previous version) to the Supplementary File (now Supplementary Figures 2-5).

The figures, plots and tables need figure captions.

Thank you. Following your suggestion, we have amended figure captions in line with Communications Psychology formatting guidelines to describe all panels and include a figure legend. These figures are now on p 14, 16, 17, and 19.

Reviewer #2 (Remarks to the Author):

This is an important, well-executed, and clearly written paper that I endorse for publication. I have only a few comments, which I hope the authors will use to further refine their already excellent manuscript.

Thank you. We are glad you enjoyed our work.

My main question pertains to the recent reversal in the gender disparity in awards—in particular, what its underlying cause(s) might be. The authors speculate that the shift can be explained in terms of changing stereotypes, with women being viewed as more competent and communal over time. This may well be a contributing factor, but it seems like a relatively small factor at best, because the shift towards parity in awards was highly nonlinear; there was a sudden, massive shift starting in 2020 that seems unlikely to (fully) reflect a gradual shift in gender stereotypes. I think it would be

worth speculating about what could have happened in recent years to cause such a drastic shift in award distribution. This is part because different underlying causes have different implications for what we can expect the future to hold. If the shift were due entirely to shifts in stereotyping, we could expect the current, relatively equitable state of affairs to persist (or even grow) in the years ahead. But if the shift we saw in 2020 was due to some acute cultural event, we might expect that progress to dissipate as time goes on. So, I think the nonlinearity of the recent shift towards equity merits further discussion.

Thank you for this suggestion. To clarify, our discussion about the impact of stereotype change over time was in respect of the Year x Award Type interaction effect showing that women are more likely to win awards for service and teaching after 1999 and 2007, and not in respect of the overall finding that women in 2020 and 2021 received more awards than men. Following your suggestion, we have added a section on p 21-22, discussing the possible influence of acute culture events including the MeToo movement and the impact of the pandemic. In respect of MeToo, this is a possible contributing factor in awards given in 2020 (which would have been nominated/decided on in late 2019, early 2020). MeToo has well documented effects in raising awareness of the persistence of gender inequality and in bringing about procedural and legal changes (see <https://www.nytimes.com/interactive/2022/10/03/us/me-too-five-years.html> and <https://www.nytimes.com/interactive/2017/11/10/us/men-accused-sexual-misconduct-weinstein.html?rref=collection%2Ftimestopic%2FWeinstein%2C%20Harvey>). One thing to note was that in 2020 women (n = 75) received a similar number of awards to the previous year i.e., 2019 (n = 76) while men's number of awards was reduced quite strikingly from 83 (in 2019) to 43 awards (in 2020). In 2021, women's awards also dropped back to 49 from 75, although women maintained a small lead on men of 8 awards. This overall drop in awards in 2021 for women and men could be explained by the impact of the pandemic. For example, many societies cancelled conferences and awards ceremonies and may have opted to skip a year (Service, 2020). Of course, these explanations are post hoc, and further research is needed to explain what is driving these trends and indeed whether they continue in the short and medium term. We have included discussion of these possible influences on p 21-22 in the discussion section:

“In 2020-2021, women even outstripped men in total awards. Such a stark change in pattern suggests the potential influence of acute cultural events. The #MeToo movement was a watershed moment for gender relations in 2017-18, which spotlighted the persistence of gender inequality in many sectors⁴⁹. While our data cannot test this, it is possible that greater collective awareness of gender bias influenced award giving in the following years. For example, there is evidence that in the wake of MeToo, the Nobel committee has explicitly asked scientists to consider gender/sex when making nominations⁵⁰. We also observed an overall drop in awards given in 2021. This may reflect the cancellation or postponement of conferences and society meetings due to the Covid-19 pandemic⁵¹. Further research should continue to track gender disparities in award giving to disentangle the influence of one-off events from longer term factors.”

A couple minor comments:

On lines 54-55, the authors distinguish descriptive from prescriptive elements of gender stereotypes. However, it's not clear what role this conceptual distinction plays in the authors' theorizing, or how it bears on the results. In fact, as far as I can tell, the descriptive/prescriptive distinction is not

mentioned at all outside that one sentence. I suggest the authors either drop that sentence or clarify how it is relevant.

Thank you for this comment. We agree that the importance of the distinction is not sufficiently clear, we have removed this sentence for clarity. Please see p 1: “Gender stereotypes describe how women and men are, distinguishing them in terms of communality (i.e., other-orientated traits) and agency (i.e., self-orientated traits¹⁵).”

The authors discuss work showing that explicit stereotyping of women has changed over the decades, but there is also work by Charlesworth and colleagues (e.g., Charlesworth et al., 2022, Patterns of Implicit and Explicit Stereotypes III: Long-Term Change in Gender Stereotypes) showing a similar pattern at the implicit level. I think that citing and discussing this work would add value. In addition, the authors might consider mentioning in the introduction that gender stereotypes have become more egalitarian over time, as this would help set the stage for the results that follow.

Thank you for these suggestions, we have now included this work from Charlesworth et al 2022 in the discussion section on p 22:

“It is also worth considering our findings in light of recent evidence showing changes in stereotyping. Research shows implicit stereotyping of men with science/career and women with arts/family has decreased between 2012 and 2018; albeit the overall tendency towards stereotypical associations remains¹⁹.”

and we have also added a sentence on p 1-2. describing the greater egalitarianism of gender stereotypes over time:

“They tend to be consensual across cultures¹⁷ and relatively stable over time¹⁸, although recent research suggests a shift towards greater gender egalitarianism at an implicit and explicit level^{18,19}.”

11th Mar 24

Dear Aífe,

Thank you for your patience during the peer-review process. Your manuscript titled "Gender Disparities in Professional Social and Personality Psychology Awards from 1968 to 2021" has now been seen by Reviewer #1. The reviewer found your revisions satisfactory and has no further requests.

We accordingly remain very interested in the possibility of publishing your study in *Communications Psychology*, but before we can make a final decision, we require you to address a methodological issue that stood out in our editorial evaluation.

The current Methods section does not provide sufficient detail on the inclusion/exclusion criteria; this pertains to how the awarding institutions were identified, how the systematic search for institutions was conducted, how comprehensive the resulting list is, and whether any institutions were omitted from the entire analysis. We ask you to revise your manuscript to provide greater presentational clarity and highlight that given the centrality of these research design choices, the revision may be subject to further external review.

I am attaching an Editorial Requests Table that details critical reporting requirements for the revised manuscript. Please attend to each item and ensure your manuscript is fully compliant. We are requesting that your manuscript aligns with these requirements as this facilitates the evaluation of your manuscript, reducing delays in re-review and potential future acceptance. If your revised manuscript is not aligned with these requests on major issues, such as those concerning statistics, it may be returned to you for further revisions without re-review. Additional information can be found in our style and formatting guide *Communications Psychology* formatting guide.

Please use the following link to submit your

- revised manuscript,
- point-by-point response to the referees' comments,
- cover letter (as a separate document),
- the Editorial Policy Checklist (see below),
- the Reporting Summary (see below), and
- the completed Editorial Request Table (attached):

[link redacted]

Best regards,

Marike

Marike Schiffer, PhD

Chief Editor

Communications Psychology

REVIEWER REPORTS:

The manuscript was re-reviewed by Reviewer #1 and the reviewer had no further comments.

EDITORIAL POLICIES

We ask that you ensure your manuscript complies with our editorial policies and reporting requirements.

To that end, we require revised manuscripts to be accompanied by two completed items: a reporting summary that collects information on study design and procedure, and an editorial policy checklist that verifies compliance with all required editorial policies.

Nature Research Reporting Summary

Editorial Policy Checklist

All points on the policy checklist must be addressed. Your revised manuscript can only be sent back to the referees if these checklists are completed and uploaded with the revision.

Notes: If you have submitted a Stage 1 Registered Report, Review, Primer, Comment, or Perspective you do not need to submit these forms. If you have already submitted these forms, you may disregard this request.

Communications Psychology Response to Editor/Reviewers' Comments

Manuscript: "Gender Disparities in Professional Social and Personality Psychology Awards from 1968 to 2021"

REVIEWERS' COMMENTS:

Reviewer 1 Comments:

The reviewer found your revisions satisfactory and has no further requests.

Thank you. We appreciate you taking the time to review our revision.

6th May 24

Dear Aife,

Your manuscript titled "Gender Disparities in Professional Social and Personality Psychology Awards from 1968 to 2021" has now been editorially evaluated. I am delighted to say that we are happy, in principle, to publish a suitably revised version in Communications Psychology under the open access CC BY license (Creative Commons Attribution v4.0 International License).

We therefore invite you to revise your paper one last time to comply with our format requirements and to maximise the accessibility and therefore the impact of your work.

EDITORIAL REQUESTS:

SUBMISSION INFORMATION:

OPEN ACCESS:

Communications Psychology is a fully open access journal. Articles are made freely accessible on publication under a CC BY license (Creative Commons Attribution 4.0 International License). This license allows maximum dissemination and re-use of open access materials and is preferred by many research funding bodies.

For further information about article processing charges, open access funding, and advice and support from Nature Research, please visit <https://www.nature.com/commspsychol/article-processing-charges>

At acceptance, you will be provided with instructions for completing this CC BY license on behalf of all authors. This grants us the necessary permissions to publish your paper. Additionally, you will be asked to declare that all required third party permissions have been obtained, and to provide billing information in order to pay the article-processing charge (APC).

* **DATA AVAILABILITY:**

[link redacted]

Best wishes,

Marike

Marike Schiffer, PhD

Chief Editor

Communications Psychology